# Inner Structure of the Lateral Geniculate Complex of Adult and Newborn *Acomys cahirinus*

**DOI:** 10.3390/ijms25147855

**Published:** 2024-07-18

**Authors:** Natalia Merkulyeva, Aleksandr Mikhalkin, Aleksandr Veshchitskii

**Affiliations:** Neuromorphology Laboratory, Pavlov Institute of Physiology of Russian Academy of Sciences, St. Petersburg 199034, Russia; mikhalkin@infran.ru (A.M.); veshchitskiiaa@infran.ru (A.V.)

**Keywords:** acomys, lateral geniculate nucleus, calbindin, calretinin, parvalbumin, GAD67, SMI-32, postnatal development

## Abstract

*Acomys cahirinus* is a unique Rodentia species with several distinctive physiological traits, such as precocial development and remarkable regenerative abilities. These characteristics render *A. cahirinus* increasingly valuable for regenerative and developmental physiology studies. Despite this, the structure and postnatal development of the central nervous system in *A. cahirinus* have been inadequately explored, with only sporadic data available. This study is the first in a series of papers addressing these gaps. Our first objective was to characterize the structure of the main visual thalamic region, the lateral geniculate complex, using several neuronal markers (including Ca^2+^-binding proteins, glutamic acid decarboxylase enzyme, and non-phosphorylated domains of heavy-chain neurofilaments) to label populations of principal neurons and interneurons in adult and newborn *A. cahirinus*. As typically found in other rodents, we identified three subdivisions in the geniculate complex: the dorsal and ventral lateral geniculate nuclei (LGNd and LGNv) and the intergeniculate leaflet (IGL). Additionally, we characterized internal diversity in the LGN nuclei. The “shell” and “core” regions of the LGNd were identified using calretinin in adults and newborns. In adults, the inner and outer parts of the LGNv were identified using calbindin, calretinin, parvalbumin, GAD67, and SMI-32, whereas in newborns, calretinin and SMI-32 were employed for this purpose. Our findings revealed more pronounced developmental changes in LGNd compared to LGNv and IGL, suggesting that LGNd is less mature at birth and more influenced by visual experience.

## 1. Introduction

The choice of animal models is crucial in physiology, cellular biology, and related disciplines. A particularly intriguing model is the rodent genus *Acomys*, or spiny mouse (specifically *Acomys cahirinus*), known for its advanced regenerative capabilities, which have been extensively studied in the context of skin and muscle regeneration [1,2,3,4].

Acomys is a precocial animal. At birth, its eyes are fully open and the animal can engage in visually guided locomotion [5]. However, limited data are available on the structure and postnatal development of the visual system in *A. cahirinus* [6]. This paper is the first in a series of publications to explore the structure and postnatal development of the central nervous system, and the visual region in particular, in *A. cahirinus*. Our first objective was to examine the main visual thalamic structure, the lateral geniculate nucleus (LGN), in adult and newborn *A. cahirinus*. In this study, we used several neurochemical markers to visualize distinct populations of interneurons and relay neurons, including Ca^2+^-binding proteins, the enzyme glutamic acid decarboxylase isoform with 67 kDa molecular weight (GAD67), and non-phosphorylated domains of high-weighted neurofilaments. In the LGN, Ca^2+^-binding proteins such as calbindin (CB), calretinin (CR), and parvalbumin (PV) are predominantly expressed in local populations of interneurons across rodents [7,8], carnivores [9,10], and primates [11,12]. The enzyme GAD67 catalyzes GABA synthesis and is found in inhibitory interneurons [13,14]. The non-phosphorylated domains of the high-weighted neurofilaments labeled by SMI-32 antibodies [15] in rodents, carnivores, and primates are predominantly expressed in the large relay geniculate neurons [7,16,17]. In carnivores and primates, SMI-32 antibodies also serve as selective markers for visual neurons involved in motion detection and spatial vision [16,18]. *A. cahirinus* exhibits strong depth perception [19,20] and relies more on visual cues compared to their closest relatives, the gerbil species [19,21,22]. Therefore, we expect to observe a well-developed population of SMI-32-immunopositive cells in the LGN of *A. cahirinus*.

## 2. Results

### 2.1. The Location of the LGN in Stereotaxic Levels

In adult *A. cahirinus*, three distinct compartments of the LGN are observable: the LGNd and LGNv nuclei, separated by the intergeniculate leaflet (IGL). At the most rostral and caudal levels, only the LGNd is present (Figure 1). For subsequent analysis, we used only slices #13–20. Similarly, in newborn *A. cahirinus*, three compartments of the geniculate complex are evident: the LGNd and LGNv nuclei and the IGL. However, in newborns, the geniculate complex was situated more rostrally compared to adults. Note that in newborns, the borders of the LGNv were more elusive. Another difference is in newborns, the optical fibers were weakly visible (Figure 2). For subsequent analysis, we used only slices #17–23.

### 2.2. Area of the Lateral Geniculate Complex

In adults, the LGNd, LGNv, and IGL nuclei areas measured 0.54 ± 0.05, 0.29 ± 0.02, and 0.04 ± 0.005 mm^2^, respectively. In newborns, these areas measured 0.29 ± 0.05, 0.21 ± 0.04, and 0.03 ± 0.005 mm^2^. Significant increases in nuclear size were observed between P0 and adults for all nuclei (LGNd: *p* < 0.0001, LGNv: *p* < 0.0001, IGL: *p* < 0.0001; Mann–Whitney test) (Figure 3A). The postnatal increase in nuclear area was 48% for the LGNd and 29% and 25% for the LGNv and IGL, respectively. The volumes of the LGNd and LGNv in adults were 0.79 mm^3^ and 0.18 mm^3^, and in P0—0.38 mm^3^ and 0.16 mm^3^. Owing to postnatal brain growth, we compared the absolute number of labeled neurons and their cellular density. However, owing to the small size of the IGL, even minor fluctuations in the number of labeled neurons can result in significant changes in cellular density. Therefore, cellular density analysis was not performed for this nucleus.

### 2.3. Calbindin Staining

*Adults.* In CB-stained slices of adult *A. cahirinus*, dark immunopositive neurons were observed throughout the LGNd, with solitary dark CB+ neurons located in the innermost part of LGNv and a moderate number in the IGL (Figure 4). CB+ neurons in the LGNd typically had oval-shaped soma without clear orientation. In the IGL, CB+ neurons had elongated somas, primarily oriented along the border between the LGNd and LGNv. In the LGNv, CB+ neurons predominantly featured oval-shaped soma with dark-stained proximal processes. No differences in shape factor were found between nuclei (LGNd vs. LGNv: *p* > 0.9999, LGNd vs. IGL: *p* = 0.4008, LGNv vs. IGL: *p* > 0.9999; Friedman test) (Table 1). Similarly, no differences in soma size were observed between nuclei (soma area: LGNd vs. LGNv: *p* = 0.4008; LGNd vs. IGL: *p* > 0.9999; IGL vs. LGNv: *p* = 0.4008; maximal diameter: LGNd vs. LGNv: *p* > 0.9999; LGNd vs. IGL: *p* = 0.6339; IGL vs. LGNv: *p* = 0.2404; minimal diameter: LGNd vs. LGNv: *p* > 0.9999; LGNd vs. IGL: *p* = 0.4008; IGL vs. LGNv: *p* > 0.9999; Friedman test) (Table 1). The LGNd nucleus exhibited a higher density of labeled neurons compared to the LGNv and IGL (*p* = 0.0022, *p* = 0.0260; Friedman test) (Table 2). The cellular density of CB+ neurons in the LGNd was more than 10 times higher compared to that in the LGNv (*p* = 0.0081; Friedman test) (Table 2). Additionally, in the outermost part of the LGNv, a distinct stripe of dark CB+ neuropil was visible (Figure 4).

*Newborns.* In CB-stained slices of all newborns, a few dark immunopositive neurons were observed in LGNv and IGL, with CB+ neurons also evident in the LGNd in two animals (Figure 4). The LGNd nucleus contained significantly fewer labeled neurons compared to the LGNv and IGL (*p* = 0.0092, *p* = 0.0128; Friedman test) (Table 2). Therefore, we analyzed data only for the LGNv and IGL. As in adults, labeled neurons have oval-shaped somas. No regional differences in shape factor were observed (*p* = 0.2783; Wilcoxon test) (Table 1). Soma size and cellular density of CB+ neurons are presented in Table 1 and Table 2. There were no differences in soma size between the LGNv and IGL nuclei (soma area: *p* = 0.7334; maximal diameter: *p* = 0.8501; minimal diameter: *p* = 0.3804; Mann–Whitney test). The innermost part of the LGNv contains thin CB+ fibers positioned parallel to its inner border (Figure 4).

Surprisingly, the soma area of CB+ neurons in the LGNv of newborns is larger compared to those in adults (*p* = 0.0159; Mann–Whitney test), with no significant change in the maximal and minimal diameters (*p* = 0.0979 and *p* = 0.0825, respectively; Mann–Whitney test). In the IGL, no differences in soma area or maximal diameter were observed (*p* = 0.2962, *p* = 0.6918, respectively; Mann–Whitney test), but the minimal diameter was significantly lower in newborns (*p* = 0.0252; Mann–Whitney test). The shape factor in newborns was similar to that in adults (LGNv: *p* = 0.3837; IGL: *p* = 0.1349; Mann–Whitney test). The number of labeled neurons in the LGNd and IGL, but not in the LGNv, of newborns was significantly lower compared to that in adults (*p* < 0.0001, *p* = 0.0295, *p* = 0.5842; Mann–Whitney test) (Figure 3B). The cellular density of CB+ neurons in the LGNv was comparable to that in adults (*p* = 0.7921; Mann–Whitney test), but the cellular density in the LGNd was significantly lower (*p* < 0.0001; Mann–Whitney test).

### 2.4. Calretinin Staining

*Adults.* In CR-stained slices of adults, dark immunopositive neurons are primarily found in the LGNv and IGL, with only a few solitary CR+ neurons visible in the LGNd (Figure 5). The differences in the number of labeled neurons between the LGNd and the LGNv and IGL were statistically significant (*p* < 0.0001, *p* = 0.0039; Friedman test). In all nuclei, CR+ neurons predominantly have oval-shaped somas. The shape factor of neurons in the LGNd was lower compared to those in the LGNv (*p* = 0.0138; Friedman test) (Table 1). Soma size and cellular density of CR+ neurons are presented in Table 1 and Table 2. No differences in soma size were observed between the nuclei (soma area: LGNd vs. LGNv: *p* > 0.9999; LGNd vs. IGL: *p* = 0.0700; IGL vs. LGNv: *p* = 0.0700; maximal diameter: LGNd vs. LGNv: *p* > 0.9999; LGNd vs. IGL: *p* = 0.7705; IGL vs. LGNv: *p* = 0.2669; minimal diameter: LGNd vs. LGNv: *p* > 0.9999; LGNd vs. IGL: *p* > 0.9999; IGL vs. LGNv: *p* > 0.9999; Friedman test). The more prominent CR-staining in the LGN is observed in the neuropil, primarily located in the outer half of the LGNd and the outer part of the LGNv (Figure 5).

*Newborns.* In CR-stained slices of newborns, moderately to weakly stained CR+ neurons are found primarily in the innermost part of the LGNv and the IGL (Figure 5). In the LGNd, labeled neurons are located in the inner half, but their number varies significantly between animals. Differences in the number of cells between the LGNv and LGNd and between the LGNd and IGL are statistically significant (*p* = 0.0024, *p* = 0.0219; Friedman test). Labeled neurons have oval-shaped somas, with the shape factor of cells in the LGNd being lower compared to the LGNv but not significantly different from the IGL (*p* = 0.0052, *p* > 0.9999; Kruskal–Wallis test). The soma size and cellular density of CR+ neurons are detailed in Table 1 and Table 2. Significant differences were observed in soma area and minimal diameter between the LGNd and IGL nuclei, but not the LGNv (soma area: *p* = 0.0417, *p* = 0.0760; minimal diameter: *p* = 0.0004, *p* = 0.0760; Friedman test). Significant differences in maximal diameter were found between the LGNd and LGNv (*p* = 0.0219, *p* = 0.0760; Friedman test). The outer half of the LGNd and the outermost part of the LGNv contain dark-stained neuropil. Throughout the LGNv and in the outer half of the LGNd, thick CR+ fibers run parallel to the curvature of the LGN (Figure 5).

The soma size of CR+ neurons in newborns did not differ from those in adults (soma area: LGNd: *p* = 0.1223; LGNv: *p* = 0.4031; IGL: *p* = 0.1083; maximal diameter: LGNd: *p* = 0.7088; LGNv: *p* = 0.1721; IGL: *p* = 0.0926; minimal diameter: LGNd: *p* = 0.0737; LGNv: *p* = 0.5458; IGL: *p* = 0.1223; Mann–Whitney test). However, significant differences in shape factor were detected (LGNd: *p* = 0.0306; LGNv: *p* = 0.0073; IGL: *p* = 0.0306; Mann–Whitney test), indicating that in newborns, CR+ neurons were more elongated compared to those in adults. The absolute number of CR+ neurons was significantly higher in newborns (LGNd: *p* < 0.0001; LGNv: *p* = 0.0002; IGL: *p* = 0.0269; Mann–Whitney test). The cellular density of the CR+ neurons in newborns was also significantly higher compared to those in adults (LGNd: *p* < 0.0001; LGNv: *p* < 0.0001; Mann–Whitney test) (Figure 3C).

### 2.5. Parvalbumin Staining

*Adults.* In PV-stained slices of adults, PV+ neurons are found only in the LGNv. These dark-stained neurons are primarily located in the middle part of the LGNv, near its outermost region (Figure 6). Labeled neurons generally exhibit oval-shaped soma. The soma size and cellular density of PV+ neurons are presented in Table 1 and Table 2. Dark PV+ neuropil is observed throughout the LGNd and in the outer half of the LGNv (Figure 6). The IGL shows no PV+ staining.

*Newborns.* In PV-stained slices of newborns, only solitary PV+ neurons were observed in the LGNv (Figure 6). Labeled neurons generally have oval-shaped soma. The soma size and cellular density of PV+ neurons are presented in Table 1 and Table 2. Dark-stained neuropil is observed throughout the LGNd. No neuropil staining is observed in the IGL and LGNv. Dark-stained fibers are primarily found in the ventral end of the LGNv (Figure 6).

The soma size of PV+ neurons in newborns did not differ from those in adults (soma area: *p* = 0.3949; maximal diameter: *p* = 0.1220; minimal diameter: *p* = 0.2743; Mann–Whitney test). However, the shape factor in newborns was significantly higher (*p* = 0.0044; Mann–Whitney test), indicating that PV+ neurons were less elongated in these animals. The absolute number of PV+ neurons and their cellular density were significantly lower in newborns (*p* < 0.0001, *p* < 0.0001; Mann–Whitney test) (Figure 3D).

### 2.6. Glutamic Acid Decarboxylase Staining

*Adults.* In GAD67-stained slices of adults, dark immunopositive neurons were predominantly observed in the LGNd, with fewer in the outer half of the LGNv (Figure 7); in 2 animals, labeled neurons were also present in the IGL. The soma size and cellular density of GAD67+ neurons are presented in Table 1 and Table 2. No statistical analysis was performed for the labeled neurons in the IGL owing to insufficient sample size. GAD67+ neurons in the LGNd and LGNv typically displayed oval-shaped soma; no inter-nuclear differences in shape factor were observed (*p* = 0.1250; Wilcoxon test). Similarly, no differences in soma size were found between LGNd and LGNv (soma area: *p* = 0.4375; maximal diameter: *p* = 0.4375; minimal diameter: *p* > 0.9999; Wilcoxon test). The number of labeled neurons was slightly higher in the LGNd compared to the LGNv (*p* = 0.0760; Friedman test) but significantly higher compared to those in the IGL (*p* < 0.0001; Friedman test). Cellular density in the LGNd was five times higher compared to that in the LGNv (*p* = 0.0020; Wilcoxon test). Throughout the LGN, except for the outermost part of the IGL, dark GAD67+ neuropil was observed (Figure 7). This neuropil staining often obscured soma staining, complicating image analysis. In the LGNv, the most prominent neuropil staining was located in the outer half.

*Newborns.* In GAD67-stained slices of newborns, immunopositive neurons were also located in the LGNd and LGNv (Figure 7). Similar to adults, labeled neurons exhibited oval-shaped soma with no inter-nuclear differences in shape factor (*p* > 0.9999; Wilcoxon test). Soma size and cellular density of GAD67+ neurons are detailed in Table 1 and Table 2. No differences in soma size were found between LGNd and LGNv (soma area: *p* = 0.3125; maximal diameter: *p* = 0.2188; minimal diameter: *p* = 0.1563; Wilcoxon test). Similar to adults, the absolute number of GAD67+ neurons in the LGNd was significantly higher compared to that in the IGL (*p* = 0.0016; Friedman test). The cellular density in the LGNd was twice as high as in the LGNv, though this difference was not statistically significant (*p* = 0.0625; Wilcoxon test). Dark-stained neuropil was observed throughout the LGN, with the LGNv displaying more prominent staining (Figure 7). Statistical comparisons related to the LGNv were omitted owing to the small sample size.

The absolute number of labeled neurons in the LGNd of adults was significantly higher than in newborns (*p* = 0.0031; Mann–Whitney test) (Figure 3E), whereas the cellular density in the LGNv was significantly higher in newborns (*p* = 0.0047; Mann–Whitney test). No differences were observed in the soma area (*p* = 0.3676; Mann–Whitney test) and maximal soma diameter of labeled neurons in the LGNd (*p* = 0.3676; Mann–Whitney test). However, the minimal diameter of soma was significantly lower in newborns (*p* = 0.0420; Mann–Whitney test). In the LGNv, the soma size of the labeled neurons in newborns was significantly smaller than that in adults (soma area: *p* = 0.0087; maximal diameter: *p* = 0.0260; minimal diameter: *p* = 0.0411; Mann–Whitney test).

### 2.7. Non-Phosphorylated Heavy-Chain Neurofilament Staining

*Adults.* In SMI-32-stained slices of adults, dark immunopositive neurons were observed in the LGNd and the outer half of the LGNv; in the IGL, only solitary labeled neurons were detected (Figure 8). LGNd contained significantly more labeled neurons compared to LGNv (*p* < 0.0001; Friedman test), and LGNv contained significantly more labeled neurons compared to IGL (*p* = 0.0138; Friedman test) (Figure 8). The soma of these neurons was mainly round; neurons in the LGNv had a higher shape factor compared to neurons in the LGNd (*p* = 0.0002; Wilcoxon test). Soma area and minimal diameters of neurons in both nuclei were similar (*p* = 0.0906 and *p* = 0.3910; Wilcoxon test); however, the maximal diameter was significantly larger in the LGNv (*p* = 0.0040; Wilcoxon test). Notably, the dark staining of the neuropil throughout the LGNd and LGNv complicated the recognition of neurons (Figure 8). The soma size and numerical density of SMI-32+ neurons are detailed in Table 1 and Table 2. Cellular density of neurons in the LGNv was significantly higher compared to that in the LGNd (*p* = 0.0107; Wilcoxon test).

*Newborns.* In SMI-32-stained slices of newborns, immunopositive neurons were observed in the LGNd and the middle part of the LGNv, near their outermost part; solitary labeled neurons were also detected in the IGL of some animals (Figure 8). Significantly more SMI-32+ neurons were found in the LGNd compared to the LGNv (*p* = 0.0001; Wilcoxon test). The same pattern was observed for cellular density (*p* = 0.0134; Wilcoxon test). Labeled neurons exhibited oval to round shapes, with significant differences in shape factor between LGNd and LGNv (*p* = 0.0002; Wilcoxon test). The soma size and cellular density of SMI-32+ neurons are detailed in Table 1 and Table 2. The soma size of SMI-32+ neurons located in the LGNv was larger compared to those in the LGNd (soma area: *p* = 0.0001; maximal diameter: *p* = 0.0001; minimal diameter: *p* = 0.0001; Wilcoxon test). Stained neuropil can mainly be observed in the inner half of the LGNd and the inner and middle parts of the LGNv, but it is absent in the outermost part (Figure 8).

In newborns, significantly more SMI-32+ neurons were found in all nuclei compared to those in adults (LGNd: *p* = 0.0127, LGNv: *p* < 0.0001, IGL: *p* = 0.0205; Mann–Whitney test) (Figure 3F). The soma size of labeled neurons in the LGNd of adults was significantly larger than that in newborns (soma area: *p* = 0.0241; maximal diameter: *p* = 0.0067; minimal diameter: *p* = 0.0122; Mann–Whitney test). The shape factors of labeled neurons in the LGNd and LGNv were similar to those in adults (*p* = 0.6673, *p* = 0.1936; Mann–Whitney test). The cellular density of SMI-32+ neurons in newborns’ LGNd and LGNv was significantly higher than of those in adults (*p* < 0.0001, *p* < 0.0001; Mann–Whitney test).

### 2.8. Comparison of Different Staining

In rodents, Ca^2+^-binding proteins mainly label interneuronal populations, whereas SMI-32 antibodies mainly label the relay cells [7]. Therefore, we compared the morphometric parameters of neurons labeled by different markers. In the LGNd of adults, the soma size of SMI-32+ neurons was significantly larger compared to those in CB+, CR+, and GAD67+ neurons (*p* = 0.0002, *p* < 0.0001, *p* = 0.0374; Kruskal–Wallis test) (Figure 9A). Similarly, in the LGNv of adults, the soma size of SMI-32+ neurons was significantly larger compared to that of CB+ and CR+ neurons (*p* = 0.0002, *p* < 0.0001; Kruskal–Wallis test) (Figure 9B). In adults, the shape factor of SMI-32+ neurons located in the LGNd was significantly higher compared to those in CB+, CR+, and GAD67+ neurons (*p* < 0.0001, *p* < 0.0001, *p* = 0.0386; Kruskal–Wallis test) (Figure 9A). Similarly, in the LGNv of adults, SMI-32+ neurons had a significantly higher shape factor compared to those in CB+ and CR+ neurons (*p* = 0.0002, *p* < 0.0001; Kruskal–Wallis test) (Figure 9B). This indicates that in adults, SMI-32+ neurons in both LGNd and LGNv are larger and have a more rounded shape compared to other labeled neurons.

In newborns, the soma size of SMI-32+ neurons in the LGNd was significantly larger compared to those in CR+ and GAD67+ neurons (*p* = 0.0099, *p* = 0.0045; Kruskal–Wallis test) (Figure 9A). Similarly, in the LGNv of newborns, the soma area of SMI-32+ neurons was significantly larger compared to those in CB+, CR+, PV+, and GAD67+ neurons (*p* = 0.0060, *p* < 0.0001, *p* = 0.0184, *p* = 0.0013; Kruskal–Wallis test). In the LGNd of newborns, the shape factor of SMI-32+ neurons was significantly higher compared to those in CB+ and CR+ neurons (*p* = 0.0006, *p* < 0.0001; Kruskal–Wallis test) (Figure 9A). Similarly, in the LGNv of newborns, the shape factor of SMI-32+ neurons was significantly higher compared to that of those in CB+ and CR+ neurons (*p* = 0.0044, *p* = 0.0131; Kruskal–Wallis test), and the shape factor of PV+ neurons was significantly higher compared to those in CB+ and CR+ neurons (*p* < 0.0001, *p* = 0.0001; Kruskal–Wallis test) (Figure 9B).

## 3. Discussion

*Morphometry of the lateral geniculate complex.* In newborn *A. cahirinus*, the volumes of LGNd and LGNv were 48.60% and 90.96% of those in adults. These data contrast with findings in immature rodents. For instance, in newborn rats [23] and gerbils [24], the LGN (LGNd) volume is one-tenth that of adults: 0.1 vs. 1.16 mm^3^ and 0.1 vs. 1.2 mm^3^, respectively. In 5-day-old mice, the volume of LGNd is one-third that of adults: 0.6 vs. 1.8 mm^3^ [25]. In our study, the total LGN volume in newborn *A. cahirinus* was 56.3% of that in adults; note that the LGNv volume was approximately the same, like in adults. This difference likely occurs because *A. cahirinus*, unlike rats, mice, and gerbils, are precocial animals.

*Structural division of the lateral geniculate complex in rodents.* In *A. cahirinus*, the three main compartments of the LGN–the LGNd, LGNv, and IGL–are observed, consistent with findings in other rodents, such as rats [26,27], mice [28], hamsters [29], gerbils [24,30], rock cavy [31], Zambian mole-rat [32], and rock hyrax [33]. Similar reports of LGNd and LGNv exist in guinea pigs [34,35], nutrias [36]), greater cane rats [37], and squirrels [35]. Among these, rats and mice have been the subjects of the most extensive LGN studies to date.

In rats and mice, the LGNd consists of the outer “shell” and the inner “core” [27,38,39]. The “shell” is a thin lamina located just beneath the optic tract, receiving axonal terminations from ON–OFF direction-selective retinal ganglion cells [38,40,41,42,43,44,45,46] and containing direction-selective W-like cells ([38,47]; we save an initial terminology when “-like” was used). Neurons in the “shell” project to layer I of the primary visual cortex [38,44]. In contrast, the “core” region of the LGNd primarily receives retinal inputs from Y (alpha) retinal ganglion cells [38,46,48] and X retinal ganglion cells [45]. The “core” region of the LGNd contains X-like and Y-like cells [47] and primarily projects to layers IV and VI of the primary visual cortex [44]. In the LGNv, retinal axons terminate mainly in the outer layer [39,43,49], forming a distinct visible “stripe”. The IGL of rats receives retinal afferents [50] and projects to multiple diencephalic structures [51]. Though the LGN of gerbils is also divided into LGNd, LGNv, and IGL [52,53], there are no reports on the “shell”, “core”, or other subdivisions in the LGNd in this species.

*Calbindin staining.* In studies involving adult rats [26,54], mice [28], and squirrels [7], most CB+ neurons in the LGNd were found in the outer part of the nucleus, specifically in the “shell” region. This finding contrasts with our data because in the LGNd of adult *A. cahirinus*, CB+ neurons were distributed throughout the entire nucleus. As for other compartments of the geniculate complex, similar to our study, CB+ neurons were found in the inner part of the LGNv and the IGL in adult rats [26] and mice [28].

In most newborn *A. cahirinus*, no CB+ neurons were found in the LGNd, although some CB+ cells were detected in the LGNv. This finding partially aligns with observations in rats, where there is a delay in the appearance of CB labeling between the LGNd and LGNv, favoring the LGNv at postnatal days P17 vs. P7 [55]. Another study in rats reported the absence of CB+ neurons at P2; labeled cells began appearing during the second postnatal week [56]. The presence of CB+ neurons in newborn *A. cahirinus* as early as at birth is consistent with these animals being precocial. We note that at birth, *A. cahirinus* exhibited mature parameters of CB-staining in the LGNv and IGL but not in the LGNd. This suggests that, unlike the LGNd, the development of these nuclei is less dependent on early sensory experiences.

*Calretinin staining.* In adult rats, CR+ neurons were predominantly found in the IGL and the inner part of the LGNv, with occasional labeling in the LGNd [8,57,58,59], which aligns with the current findings in adult *A. cahirinus*. Similar to our study, dense bundles of labeled fibers were observed primarily in the outer parts of the LGNd [57,58,59] and LGNv [8,57,58,59].

In rats, the first CR+ neurons in the LGNv and IGL were detected as early as embryonic day 19 [60], suggesting CR as a marker for early generated thalamic cells. Newborn *A. cahirinus* had more CR-positive neurons in all nuclei of the geniculate complex, particularly in the IGL and LGNv, compared to adults. This may indicate that CR+ networks in newborns organize transient populations, similar to findings in the perigeniculate nucleus of cats [61].

*Parvalbumin staining.* The PV-labeling of the LGN in adult *A. cahirinus* resembles that observed in adult rats [50,62,63]: soma staining is prominent in the LGNv, with dense neuropil staining in the LGNd and the outer part of the LGNv. It was also noted that soma labeling is primarily located in the middle part of the LGNv [63]. However, no PV+ somas were observed in either the LGNd or LGNv in [64]. Previously, dense PV+ fiber bundles combined with very rare soma staining throughout the LGNd were also observed in rats [54,64], as noted in the study. In squirrels, PV-labeling is predominantly found in the outer parts of the LGNd [7], which differs from our findings. The numerical density of PV+ neurons in the LGNv of adult *A. cahirinus* is similar to that observed in rats [63]; however, unlike rats, almost no soma labeling was detected in the LGNd.

In newborn *A. cahirinus*, only single PV+ neurons in the LGNv and dark-stained neuropil in the LGNd were observed. Similarly, no PV+ neuronal somas were detected in newborn rats in the LGNd and LGNv [55,56]. These findings align with the observations in rats, indicating that the system of PV+ neurons in *A. cahirinus*, like in rats, is experience-dependent.

*GAD staining.* In adult rats, mice, and gerbils, neurons immunopositive for GAD were found in LGNd and LGNv [50,65,66,67]. Similarly, in adult *A. cahirinus*, most GAD67+ neurons were observed in the LGNd, with only solitary GAD67+ neurons found in the LGNv. More intense GAD staining was observed in the LGNv compared to the LGNd in adult mice [49,67], rats [50,68], and Nile grass rats [68], which aligns with our findings. In some studies [68], GAD65 rather than GAD67 was used. A previous study [50] showed that neuropil labeling tended to obscure soma staining, similar to what we found in our study. Consistent with our data, they reported stronger neuropil GAD staining in the outer part of the LGNv compared to the inner part in rats [69] and mice [67]. Similar differences in the staining of the outer and inner parts of the LGNv were observed in rats when GABA labeling was analyzed [70]. The cellular density in our study is several times lower than previously reported [71].

We observed GAD67 labeling in the LGN of newborn *A. cahirinus*. Similarly, GABA-immunoreactive neurons are present at birth in rats’ LGN [72]. In rats, GAD activity in the LGN increases from birth [73], as reported in this study.

*SMI-32 staining.* We did not find any data related to SMI-32 staining in the LGN of rats, mice, or gerbils. However, data on SMI-32 staining in the LGN of adult squirrels show a uniform distribution, similar to our findings in adults [7]. Our data indicate that SMI-32+ neurons have a larger soma area than CB+, CR+, and GAD67+ cells, which suggests that SMI-32+ neurons are relay cells rather than interneurons, consistent with previous findings in cats [16] and primates [74]. Additionally, we did not find any data on the postnatal development of SMI-32 staining in the LGN of rodents. However, our data on cats revealed that geniculate compartments in newborns are more clearly evident than in adults [75]; similar results were obtained for *A. cahirinus*.

SMI-32 is commonly used as a marker for the Y visual channel, responsible for motion detection [16,18,76,77,78], and an indicator of neuronal maturation (see [79]). Therefore, the specific features of SMI-32 staining in the LGN of newborns may be related to the distinct developmental characteristics of Y cells.

## 4. Materials and Methods

### 4.1. Subjects

Care and experimental procedures were carried out in accordance with the requirements of Council Directive 2010/63EU of the European Parliament on the protection of animals used in experimental and other scientific purposes and the guidelines of the National Institute of Health Guide for the Care and Use of Laboratory Animals, and with the approval of the Ethics Commission of the Pavlov Institute of Physiology (Protocol # 02/17; 17.02.2022). Two age groups of *Acomys cahirinus* of both sexes were used: (i) newborns (n = 8) and (ii) adults aged 2–3 (n = 2) and 13–16 months (n = 6). According to [6,80], olfactory bulbs and visual cortex of acomys aged 2–3 months shared adult-like features. Thereby, the data from the 2–3- and 13–16-month-old animals was combined. All animals were bred in the laboratory of neuromorphology of the Pavlov Institute of Physiology Russian Academy of Sciences, housed 3–5 per cage, and maintained under standard lab conditions (12 h light/dark cycle) with food and water provided according to the diet elaborated; all breeding conditions were presented in [81]. The body weight of newborns was 5.52–6.51 g, and of adults: 25–30 g and 44–56 g, which corresponds well to [82].

### 4.2. Perfusion and Histological Slices Preparing

Animals were deeply anesthetized with a mixture of Zoletil (Virbac, Carros, France; 20 mg/kg) and Xyla (Interchemie werken “De Adelaar” BV, Venray, Netherlands; 2 mg/kg; i/p) and, thereafter, were transcardially perfused with 0.9% NaCl (25 mL for adults and 15 mL for newborns) followed by 4% paraformaldehyde (70 mL for adults and 30 mL for newborns). After perfusion, the brains were removed from the skull and stored in 20–30% sucrose. Thereafter, the brains were cut into 40 μm frontal slices on a freezing microtome (Reichert, Vienna, Austria).

### 4.3. Immunohistochemistry

For immunohistochemical staining, slices were processed as free floating. Antigens were unmasked in 1% NaBH_4_ for 15 min (Serva Feinbiochemica, Heidelberg, Germany). Endogenous peroxidase activity was blocked by 0.3% H_2_O_2_ for 30 min, unspecific immunoreactivity was lowered by incubation in 3% bovine serum albumin for 1 h (BSA, Biolot, St. Petersburg, Russia). Slices were incubated with the primary antibodies (Table 3) for 70 h. Thereafter, they were incubated with the secondary biotinylated horse anti-mouse IgG antibodies (Vector Laboratories, Inc., Newark, NJ, USA, BA-2000, RRID: AB_2313581; dilution 1:600) or goat anti-rabbit IgG antibodies (Vector Laboratories, Inc., Newark, NJ, USA, BA-1000, RRID: AB_2313606; dilution 1:600) for 24 h. Then, slices were processed in an avidin–biotin horseradish–peroxidase complex (ABC Elite system, Vector Laboratories, Inc., Newark, NJ, USA) and, thereafter, in the diaminobenzidine-(NH_4_)_2_Ni(SO_4_)_2_·6xH_2_O-solution, then washed, mounted, dehydrated, cleared, and placed under coverslips in Bio Mount HM (Bio-Optica, Milan, Italy).

### 4.4. Image Processing

Images of the brain slices were acquired using a computer setup equipped with an Olympus CX31 light microscope (Olympus Corporation, Tokyo, Japan, 10× objective), VideoZavr Scan 2.4 software package (VideoZavr, St. Petersburg, Russia), and camera VideoZavr Standart VZ-18C23-B (VideoZavr, St. Petersburg, Russia). The divisions of the brain structures were based upon two mouse brain atlases [83,84] and the differences in immunostaining and in the Nissl staining. For every animal, three slices of the brain were taken into analysis. The image processing and all measurements (the neuronal soma area, maximal and minimal diameters of soma, soma perimeter, and area of the lateral geniculate complex nuclei) were processed using the free Fiji ImageJ software (v. 2.14.0/1.54f) [85]. The soma shape of labeled neurons was determined using the shape factor by the following formula: (4π × soma area)/(soma perimeter)^2^, in accordance with [65]. Shape factors ranged between 0 and 1, with 1 representing a perfect circle and factors less than 1 representing progressively greater degrees of ellipses; values above 0.8 mean a round shape of the soma, and those below 0.8, an oval shape.

An area of the nuclei was assessed at slices where nuclear borders were clearly detected. This was possible using calbindin-, calretinin-, GAD67-, and SMI-32-staining. The total data was averaged. The volume of the dorsal (LGNd) and ventral (LGNv) nuclei was assessed using the serial Nissl-stained frontal slices of two animals aged 0 days and 11 months, respectively; an area and the slice thickness were multiplied.

### 4.5. Statistics

Data are presented as mean ± SD. One half of the slice (one hemisphere) was used as “N”. For the multiple paired intra-group comparisons, the Friedman test with Dunn’s multiple comparisons correction was used. For the multiple non-paired intra-group comparisons, the Kruskal–Wallis test was used. For the dual paired intra-group comparisons, the Wilcoxon test was used. For the unpaired inter-group comparisons, the Mann–Whitney test was used.

## 5. Conclusions

This study is the first to evaluate the organization and development of the LGN in *A. cahirinus*. Using several neurochemical markers, we identified the following functional subdivisions of the LGN in adult and newborn *A. cahirinus*: the “shell” and “core” parts of the LGNd, the inner and outer compartments of the LGNv, and the IGL. In adults and newborns, the “shell–core” distinction of the LGNd is visible only with CR staining. In adults, differentiating the LGNv into inner and outer parts was possible using various markers, including CB, CR, PV, GAD67, and SMI-32. In newborns, this division was achievable using CR and SMI-32 staining. In newborns, the LGNv is more developed compared to the LGNd, but in general, despite their early development, the visual system of newborn *A. cahirinus* is far from mature.

### Limitations

No gender differences were investigated owing to the limited sample size. However, data about peculiarities of the placental development of male and female acomyses were obtained [86].

## Figures and Tables

**Figure 1 ijms-25-07855-f001:**
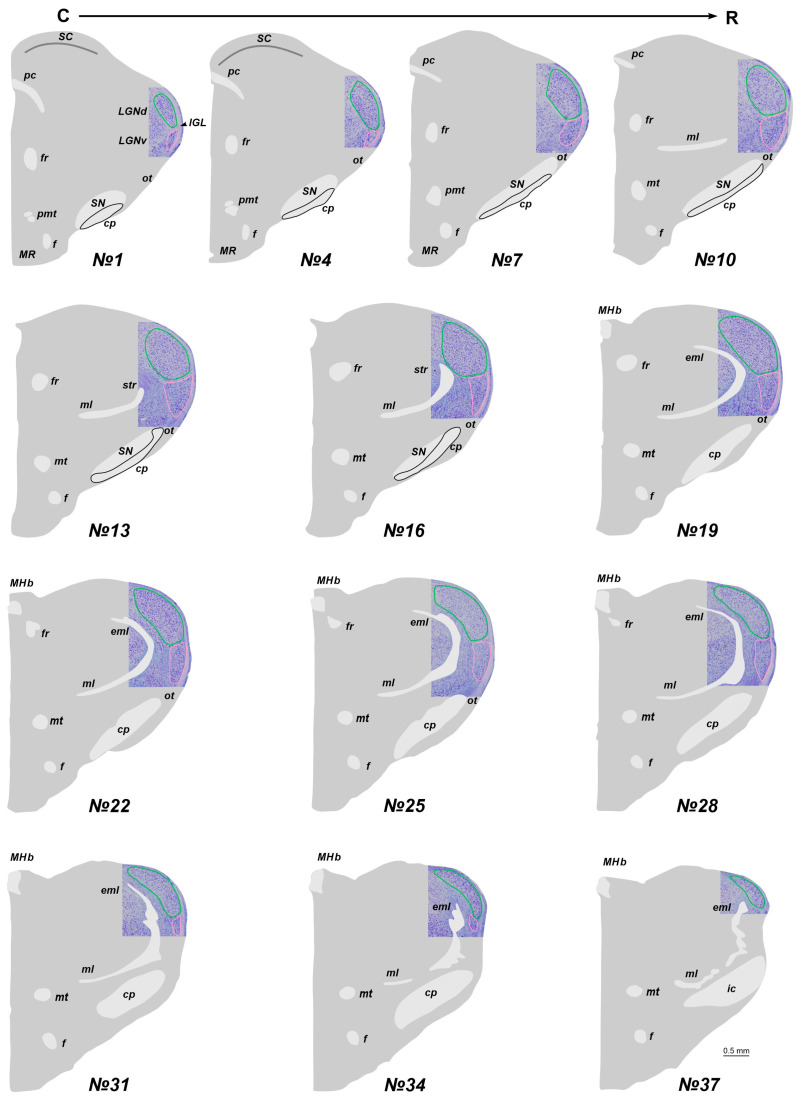
The total scheme of the lateral geniculate nucleus (LGN) location in adult acomys. The region containing LGN at the Nissl staining slice is introduced into the scheme. Numerals are numbers of the frontal slices. The dorsal and ventral nuclei of the LGN (LGNd and LGNv) are labeled by green and pink. An intergeniculate leaflet (IGL) at all levels is located between the LGNd and LGNv. cp—cerebral pedunculi; eml—external medullary lamina; f—fornix; fr—fasciculus retroflexus; ic—internal capsule; MHb—medial habenula; ml—medial lemniscus; MR—mammillary region; mt—mammillary tract; ot—optic tract; pc—posterior commissure; pmt—principal mammillary tract; SC—superior colliculus; SN—substantia nigra; str—superior thalamic radiation. C-R—caudo-rostral direction.

**Figure 2 ijms-25-07855-f002:**
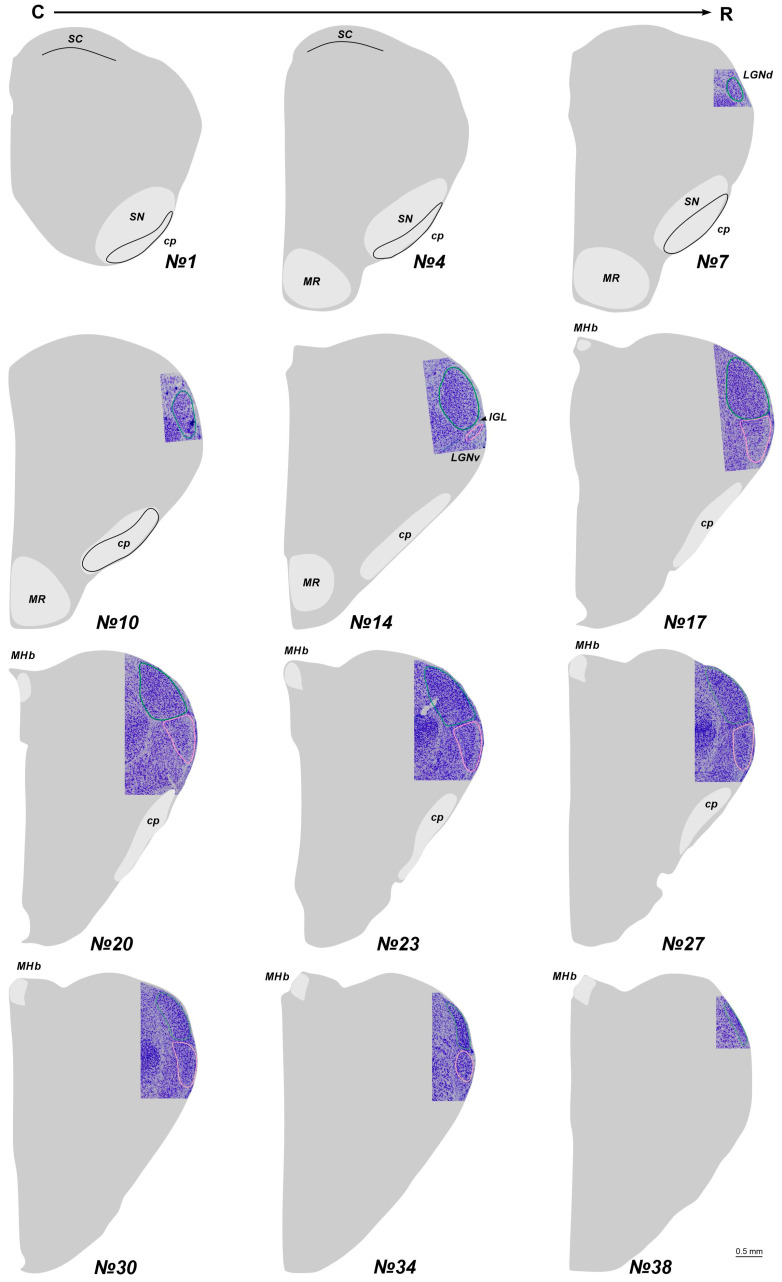
The total scheme of the lateral geniculate nucleus (LGN) location in newborn acomys. The region containing LGN at the Nissl staining slice is introduced into the scheme. Numerals are numbers of the frontal slices. The dorsal and ventral nuclei of the LGN (LGNd and LGNv) are labeled by green and pink. An intergeniculate leaflet (IGL) at all levels is located between the LGNd and LGNv. cp—cerebral pedunculi; MHb—medial habenula; MR—mammillary region; SC—superior colliculus; SN—substantia nigra; C-R—caudo-rostral direction.

**Figure 3 ijms-25-07855-f003:**
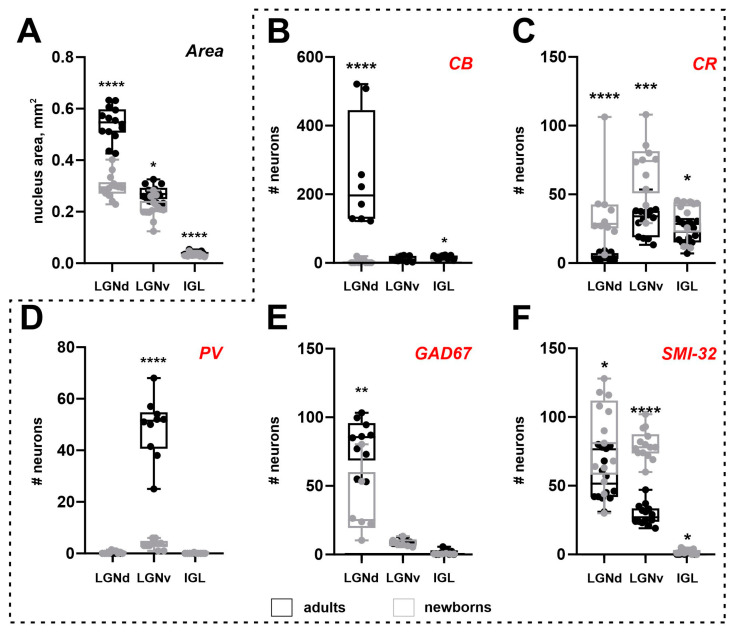
An area of nuclei of the lateral geniculate complex: dorsal lateral geniculate nucleus (LGNd), ventral lateral geniculate nucleus (LGNv), and intergeniculate leaflet (IGL) (**A**) and the number of neurons stained for the calbindin (CB), calretinin (CR), parvalbumin (PV), glutamic acid decarboxylase isoform with molecular weights of 67 kDa (GAD67), and non-phosphorylated domains of the high-weighted neurofilaments (SMI-32) of adult (black) and newborn (gray) acomys (**B**–**F**). **** *p* < 0.0001, *** *p* < 0.001, ** *p* < 0.01, * *p* < 0.05.

**Figure 4 ijms-25-07855-f004:**
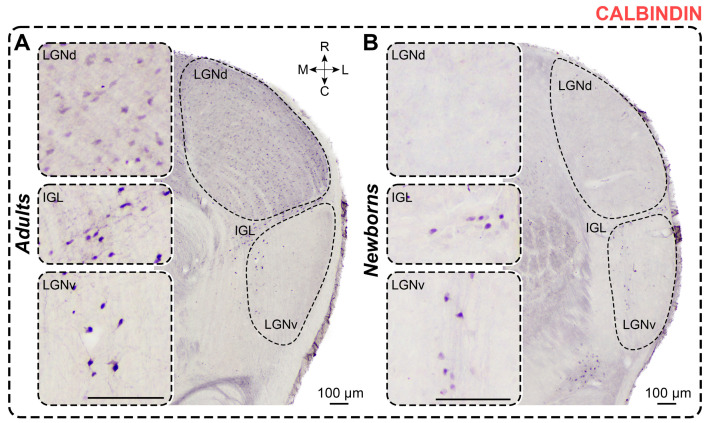
Neurochemical staining of the lateral geniculate complex in adult (**A**) and newborn (**B**) acomyses using antibodies to calbindin (CB). LGNd and LGNv—dorsal and ventral lateral geniculate nucleus, IGL—intergeniculate leaflet; R, C, M, L—rostral, caudal, medial, and lateral directions. Calibration bar is 100 µm.

**Figure 5 ijms-25-07855-f005:**
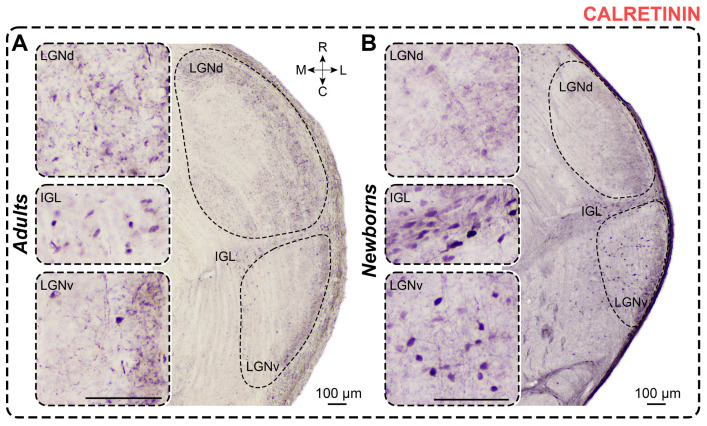
Neurochemical staining of the lateral geniculate complex in adult (**A**) and newborn (**B**) acomyses using antibodies to calretinin (CR). LGNd and LGNv—dorsal and ventral lateral geniculate nucleus, IGL—intergeniculate leaflet; R, C, M, L—rostral, caudal, medial, and lateral directions. Calibration bar is 100 µm.

**Figure 6 ijms-25-07855-f006:**
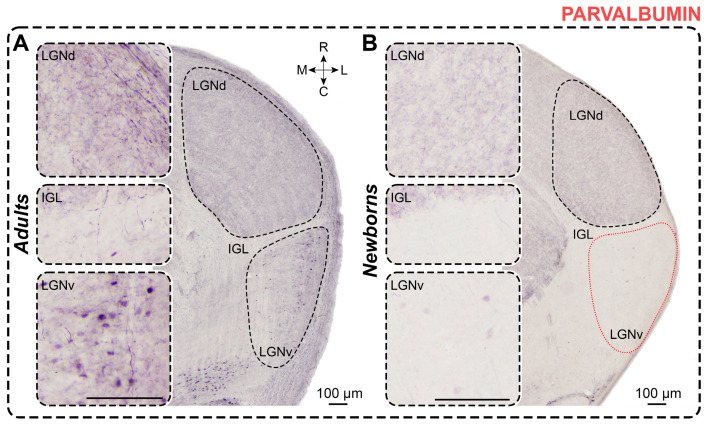
Neurochemical staining of the lateral geniculate complex in adult (**A**) and newborn (**B**) acomyses using antibodies to parvalbumin (PV). LGNd and LGNv—dorsal and ventral lateral geniculate nucleus, IGL—intergeniculate leaflet; red dotted line—the assumed boundaries of the LGNv; R, C, M, L—rostral, caudal, medial, and lateral directions. Calibration bar is 100 µm.

**Figure 7 ijms-25-07855-f007:**
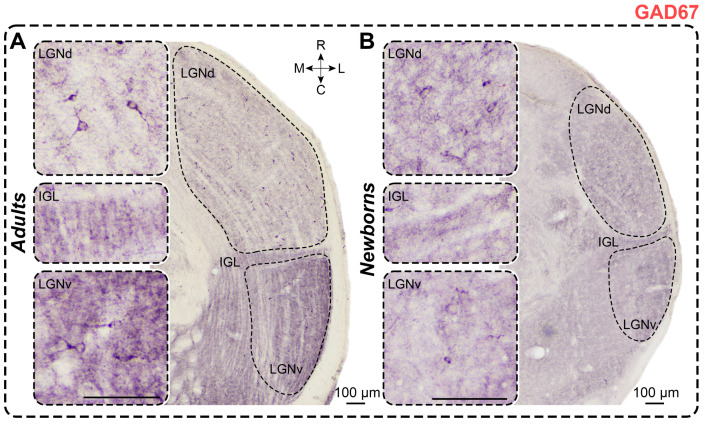
Neurochemical staining of the lateral geniculate complex in adult (**A**) and newborn (**B**) acomyses using antibodies to GAD67 (CB). LGNd and LGNv—dorsal and ventral lateral geniculate nucleus, IGL—intergeniculate leaflet; R, C, M, L—rostral, caudal, medial, and lateral directions. Calibration bar is 100 µm.

**Figure 8 ijms-25-07855-f008:**
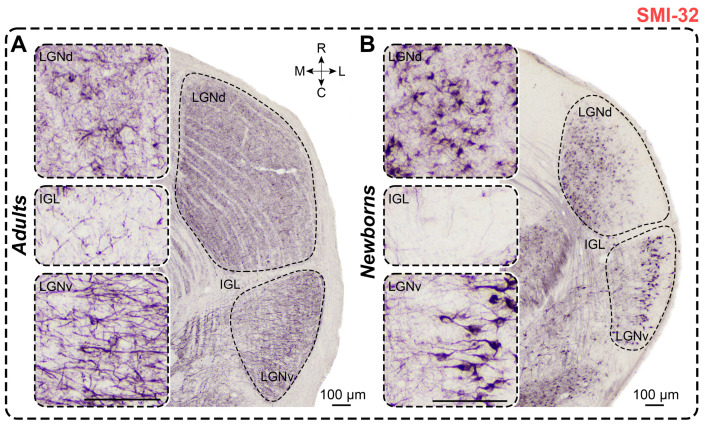
Neurochemical staining of the lateral geniculate complex in adult (**A**) and newborn (**B**) acomyses, using antibodies to SMI-32. LGNd and LGNv—dorsal and ventral lateral geniculate nucleus, IGL—intergeniculate leaflet; R, C, M, L—rostral, caudal, medial, and lateral directions. Calibration bar is 100 µm.

**Figure 9 ijms-25-07855-f009:**
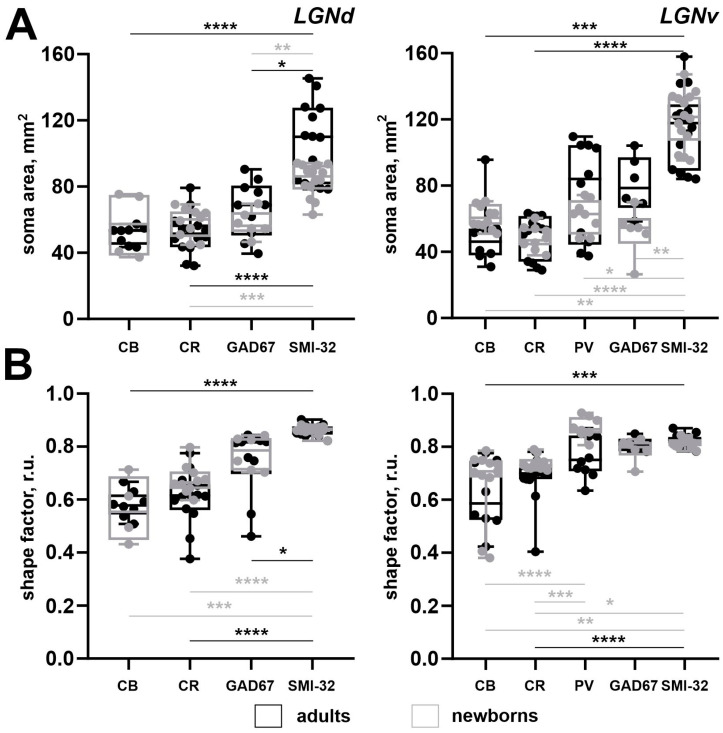
A soma area (**A**) and the shape factor (**B**) of neurons stained for the calbindin (CB), calretinin (CR), parvalbumin (PV), glutamic acid decarboxylase isoform with molecular weights of 67 kDa (GAD67, GAD), and non-phosphorylated domains of the high-weighted neurofilaments (SMI-32) of the dorsal and ventral lateral geniculate nuclei (LGNd and LGNv) of adult (black) and newborn (gray) acomys. **** *p* < 0.0001, *** *p* < 0.001, ** *p* < 0.01, * *p* < 0.05.

**Table 1 ijms-25-07855-t001:** Soma size: soma area (µm^2^) and maximal and minimal diameters (µm), and the shape factor of immunopositive neurons located in the lateral geniculate nuclei: dorsal (LGNd) and ventral (LGNv), and in the intergeniculate leaflet (IGL), of adult and newborn acomyses. CB—calbindin, CR—calretinin, PV—parvalbumin, GAD67—glutamic acid decarboxylase, SMI-32—antibody to the non-phosphorylated heavy-chain neurofilaments. Data presented as mean ± SD, Mann–Whitney test was used. ** *p* < 0.01, * *p* < 0.05, ns—non-significant.

	Adults			Newborns		
	*LGNd*	*LGNv*	*IGL*	*LGNd*	*LGNv*	*IGL*
***Soma area,*** **µm^2^**						
** *CB* **	50.9 ± 5.3	50.5 ± 20.1	53.1 ± 11.0	56.6 ± 20.6 ns	61.5 ± 6.9 *	59.6 ± 7.6 ns
** *CR* **	51.7 ± 11.6	49.9 ± 12.6	–	57.8 ± 8.7 ns	48.6 ± 6.9 ns	47.6 ± 4.9 ns
** *PV* **	–	76.4 ± 29.7	–	–	61.8 ± 10.0 ns	–
** *GAD67* **	66.8 ± 16.9	80.6 ± 17.1	–	57.5 ± 7.9 ns	52.5 ± 14.2 **	–
** *SMI-32* **	106.7 ± 23.8	115.1 ± 23.1	–	83.7 ± 9.7 *	120.8 ± 16.2 ns	–
***Maximal diameter,*** **µm**						
** *CB* **	11.8 ± 0.8	11.4 ± 2.0	12.6 ± 1.5	11.8 ± 2.4 ns	12.3 ± 1.5 ns	12.4 ± 0.9 ns
** *CR* **	11.6 ± 1.9	11.2 ± 1.6	11.9 ± 1.7	11.6 ± 1.0 ns	10.6 ± 0.8 ns	11.0 ± 0.6 ns
** *PV* **	–	12.9 ± 2.3	–	–	11.2 ± 1.2 ns	–
** *GAD67* **	12.7 ± 1.1	13.7 ± 1.4	–	12.0 ± 1.1 ns	11.2 ± 1.6 *	–
** *SMI-32* **	14.7 ± 1.5	16.1 ± 1.7	–	13.2 ± 0.7 **	16.8 ± 1.3 ns	–
***Minimal diameter,*** **µm**						
** *CB* **	7.3 ± 0.2	7.3 ± 1.7	6.9 ± 0.9	7.9 ± 1.8 ns	8.1 ± 0.9 ns	7.7 ± 0.7 *
** *CR* **	7.3 ± 0.9	7.2 ± 1.2	7.1 ± 1.2	8.0 ± 0.7 ns	7.1 ± 0.5 ns	6.7 ± 0.3 ns
** *PV* **	–	8.3 ± 1.5	–	–	7.4 ± 0.6 ns	–
** *GAD67* **	7.7 ± 0.9	8.2 ± 1.3	–	6.9 ± 0.5 *	6.6 ± 0.9 *	–
** *SMI-32* **	9.9 ± 1.1	10.0 ± 0.9	–	8.7 ± 0.6 *	10.2 ± 0.7 ns	–
***Shape factor,*** **r.u.**						
** *CB* **	0.58 ± 0.05	0.61 ± 0.13	0.62 ± 0.05	0.56 ± 0.12 ns	0.66 ± 0.14 ns	0.65 ± 0.05 ns
** *CR* **	0.60 ± 0.10	0.68 ± 0.09	0.67 ± 0.03	0.68 ± 0.05 *	0.74 ± 0.03 **	0.70 ± 0.02 *
** *PV* **	–	0.77 ± 0.08	–	–	0.87 ± 0.04 **	–
** *GAD67* **	0.75 ± 0.13	0.81 ± 0.03	–	0.78 ± 0.06 ns	0.79 ± 0.04 ns	–
** *SMI-32* **	0.86 ± 0.02	0.83 ± 0.02	–	0.87 ± 0.03 ns	0.82 ± 0.02 ns	–

**Table 2 ijms-25-07855-t002:** Numerical density (cells/mm^2^) of immunopositive neurons located in the lateral geniculate nuclei: dorsal (LGNd) and ventral (LGNv), and in the intergeniculate leaflet (IGL) of adult and newborn acomyses. CB—calbindin, CR—calretinin, PV—parvalbumin, GAD67—isoform of glutamic acid decarboxylase, SMI-32—antibody to the non-phosphorylated heavy-chain neurofilaments. Data presented as mean ± SD, Mann–Whitney test was used. **** *p* < 0.0001, *** *p* < 0.001, ** *p* < 0.01, * *p* < 0.05, ns—non-significant.

	Adults			Newborns		
	*LGNd*	*LGNv*	*IGL*	*LGNd*	*LGNv*	*IGL*
** *Number of neurons* **						
** *CB* **	257 ± 166	12 ± 8	17 ± 5	3 ± 6 ****	12 ± 4 ns	12 ± 4 *
** *CR* **	5 ± 2	31 ± 11	22 ± 8	37 ± 27 ****	69 ± 23 ***	35 ± 14 *
** *PV* **	–	49 ± 12	–	–	4 ± 2 ****	–
** *GAD67* **	81 ± 17	9 ± 2	2 ± 2	36 ± 26 **	8 ± 3 ns	0.1 ± 0.2 ns
** *SMI-32* **	56 ± 17	29 ± 8	0.8 ± 0.9	82 ± 31 *	76 ± 8 ****	2 ± 2 *
** *Cellular density* **						
** *CB* **	517 ± 394	40 ± 28	–	10 ± 17 ****	43 ± 14 ns	–
** *CR* **	10 ± 5	99 ± 32	–	124 ± 81 ****	349 ± 109 ****	–
** *PV* **	–	241 ± 74	–	–	1 ± 2 ****	–
** *GAD67* **	154 ± 36	28 ± 11	–	118 ± 85 ns	52 ± 16 **	–
** *SMI-32* **	107 ± 36	147 ± 40	–	330 ± 128 ****	492 ± 90 ****	–

**Table 3 ijms-25-07855-t003:** The list of antibodies.

Antibody	Host	Clonality	Dilution	Manufacturer	Lot	RRID
Calbindin	Mouse	Monoclonal	1:10,000	Sigma-Aldrich (Burlington, MA, USA)	C9848	AB_476894
Calretinin	Rabbit	Polyclonal	1:10,000	Sigma-Aldrich (Burlington, MA, USA)	AB5054	AB_2068506
Parvalbumin	Rabbit	Polyclonal	1:5000	Abcam (Cambridge, UK)	ab11427	AB_298032
GAD67	Mouse	Monoclonal	1:2000	Sigma-Aldrich (Burlington, MA, USA)	MAB5406	AB_2278725
SMI-32	Mouse	Monoclonal	1:5000	BioLegend (San Diego, CA, USA)	801701	AB_2564642

## Data Availability

The datasets generated during and/or analyzed during the current study are available from the corresponding author on reasonable request.

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
