# Peer review of "Inner Structure of the Lateral Geniculate Complex of Adult and Newborn Acomys cahirinus"

_ijms, 2024, doi:10.3390/ijms25147855_

Round 1

Reviewer 1 Report

Comments and Suggestions for Authors

This manuscript describes the constitution of the lateral geniculate complex (LGC), part of the auditory trajectory, in the unusual rodent, the spiny mouse, aconmys cahirinus. This mouse differentiates from many other rodents in that it has a high neurodegenerative potential, even postnatally. In the current manuscript, the LGC is studied with emphasis on stereological counting of the neuronal number both in adulthood and in the newborn. As such, the study provides novelty in that the CNS of this species is very understudied. The neurons are identified by means of their content of a set of neuronal markers, i.e. Ca2+-binding proteins (calbindin, calretinin, parvalbumin), glutamic acid decarboxylase enzyme (GAD67), and non-phosphorylated domains of heavy-chain neurofilaments (SMI-32). Not so surprisingly, a clear developmental profile is recognized during postnatal development. An interesting aspect of the study is the obvious comparative, yet different, profile of the neurons of the various parts of the LGC compared to other more well-known rodents.

Comments:
The protocol is well-described and the illustrations convincing. A hesitation lies on the number of involved animals, especially in the 2 months, but also in older animals. Please explain the decision not to include more animals.

This animal is several times in the manuscript praised for its neurodegenerative potential. However, this is not included very much in the discussion. It would of course have been of interest to see how the LGCs would react to pathology induced to the visual system. Perhaps the authors emphasize more clearly how this model might have advantages for study of visual pathways, e.g. if subjected to surgical lesion of the optic pathways during development and in the intact brain.

Author Response

Reviewer 1

This manuscript describes the constitution of the lateral geniculate complex (LGC), part of the auditory trajectory, in the unusual rodent, the spiny mouse, aconmys cahirinus. This mouse differentiates from many other rodents in that it has a high neurodegenerative potential, even postnatally. In the current manuscript, the LGC is studied with emphasis on stereological counting of the neuronal number both in adulthood and in the newborn. As such, the study provides novelty in that the CNS of this species is very understudied. The neurons are identified by means of their content of a set of neuronal markers, i.e. Ca2+-binding proteins (calbindin, calretinin, parvalbumin), glutamic acid decarboxylase enzyme (GAD67), and non-phosphorylated domains of heavy-chain neurofilaments (SMI-32). Not so surprisingly, a clear developmental profile is recognized during postnatal development. An interesting aspect of the study is the obvious comparative, yet different, profile of the neurons of the various parts of the LGC compared to other more well-known rodents.

We thank the Reviewer for the positive feedback.

Comments:

The protocol is well-described and the illustrations convincing. A hesitation lies on the number of involved animals, especially in the 2 months, but also in older animals. Please explain the decision not to include more animals.

According to the studies (Brunjes, 1983, 1985), olfactory bulbs and visual cortex in 2-3 month-old acomyses are adult-like. Unfortunately, no other data about developmental status of acomys’s brain at different ages exist. We also did not notice clear differences in the patterns of used neurochemical markers between 2-3 and 13-16 month-old animals. Therefore, they were combined into one group of adults.

We have added the following into the main text: “According to (Brunjes, 1983, 1985), olfactory bulbs and visual cortex of acomys aged 2-3 months shared adult-like features. Thereby, the data from 2-3 and 13-16 month-old animals was combined”.

Brunjes PC. Olfactory bulb maturation in Acomys cahirinus: is neural growth similar in precocial and altricial murids? Brain Res. 1983 Jun;284(2-3):335-41. doi: 10.1016/0165-3806(83)90016-0.

Brunjes PC. A stereological study of neocortical maturation in the precocial mouse, Acomys cahirinus. Brain Res. 1985 Apr;351(2):279-87. doi: 10.1016/0165-3806(85)90199-3. PMID: 3995352.

This animal is several times in the manuscript praised for its neurodegenerative potential. However, this is not included very much in the discussion. It would of course have been of interest to see how the LGCs would react to pathology induced to the visual system. Perhaps the authors emphasize more clearly how this model might have advantages for study of visual pathways, e.g. if subjected to surgical lesion of the optic pathways during development and in the intact brain.

Several opportunities to use acomys in the studies of visual system exist. (1) Using precocial animals allows us to compare the environment-dependent and independent features of the developing visual system. Previously, using a cat model, we obtained several transient neuronal populations of the LGN in newborns; we supposed they are totally independent upon the visual experience. Several similar features we have revealed in acomys, and thereby, this data supported our early supposition. (2) Due to the high regenerative potential, it allows us to use some invasive techniques (surgery implantation, tracing studies, etc) that are harder for the compatible in size routine rodents, especially, for newborns. (3) Since acomyses are born with fully opened eyes and are able to visual guided locomotion, it is possible to use this animal model for the investigation of sensory-motor integration at all postnatal studies. The combination of the 2nd and 3rd issues allows us to merge several techniques like behavioral testing at trail, treadmill, and ladder with invasive procedures just after acomys’s birth. Now we are elaborating some methodological approaches in this sense.

Reviewer 2 Report

Comments and Suggestions for Authors

In the manuscript submitted for review, the Authors analyzed inner structure of the lateral geniculate complex of adult and newborn Acomys cahirinus.

I find the topic of the manuscript interesting  and the whole work is thoughtful. The Authors put a lot of work into preparing this interesting work. The reader's attention is undoubtedly drawn to carefully prepared figures and photos. 

However, I have a few comments:

1. Were there any differences between male and female? The question arises because the Authors have studied newborns and adult animals in their experience, but what about gender?

2. The Authors wrote that “Acomys is a precocial animal. After birth, its eyes are fully open and the animal can they engage in visually guided locomotion [5]. However, limited data are available on the structure and postnatal development of the visual system in A. cahirinus [6]. This work is the first in a series of publications devoted to the study of postnatal structure and development the central nervous system, and in particular the organ of vision, in A. cahirinus.…” Why was this species chosen? I understand that a given structure has not yet been described in this mouse, but what impact does it have on science/medicine/practice? Can the knowledge about the geniculate nucleus in Acomys be somehow transferred to higher animals (Primates) or humans?

3. Were the sections heat treated to unblock antigen?

4. "For immunohistochemical staining, slices were processed as free floating. Endogenous peroxidase activity was blocked by 0.3% H2O2, unspecific immunoreactivity was lowered by incubation in 3% bovine serum albumin...- how long?

Author Response

Reviewer 2

In the manuscript submitted for review, the Authors analyzed inner structure of the lateral geniculate complex of adult and newborn Acomys cahirinus.

I find the topic of the manuscript interesting  and the whole work is thoughtful. The Authors put a lot of work into preparing this interesting work. The reader's attention is undoubtedly drawn to carefully prepared figures and photos.

We thank the Reviewer for the positive feedback.

However, I have a few comments:

  1. Were there any differences between male and female? The question arises because the Authors have studied newborns and adult animals in their experience, but what about gender?

This work is the first to describe the structure of the geniculate neuronal populations of acomys. Since acomys is a precocial animal, the first point was to reveal potential differences between newborns and adults. Searching for the gender differences is the second point of our interest, moreover, some data about peculiarities of the placental development of male and female acomyses was obtained (O’Connell et al., 2013).

O'Connell BA, Moritz KM, Walker DW, Dickinson H. Sexually dimorphic placental development throughout gestation in the spiny mouse (Acomys cahirinus). Placenta. 2013 Feb;34(2):119-26. doi: 10.1016/j.placenta.2012.11.009.

We have added the following into the text of Limitations: “But data about peculiarities of the placental development of male and female acomyses was obtained (O’Connell et al., 2013).”.

  1. The Authors wrote that “Acomys is a precocial animal. After birth, its eyes are fully open and the animal can they engage in visually guided locomotion [5]. However, limited data are available on the structure and postnatal development of the visual system in A. cahirinus [6]. This work is the first in a series of publications devoted to the study of postnatal structure and development the central nervous system, and in particular the organ of vision, in A. cahirinus.…” Why was this species chosen? I understand that a given structure has not yet been described in this mouse, but what impact does it have on science/medicine/practice? Can the knowledge about the geniculate nucleus in Acomys be somehow transferred to higher animals (Primates) or humans?

Several opportunities to use acomys in the studies of visual system exist. (1) Using precocial animals allows us to compare the environment-dependent and independent features of the developing visual system. Previously, using a cat model, we obtained several transient neuronal populations of the LGN in newborns; we supposed they are totally independent upon the visual experience. Several similar features we have revealed in acomys, and thereby, this data supported our early supposition. (2) Due to the high regenerative potential, it allows us to use some invasive techniques (surgery implantation, tracing studies, etc) that are harder for the compatible in size routine rodents, especially, for newborns. (3) Since acomyses are born with fully opened eyes and are able to visual guided locomotion, it is possible to use this animal model for the investigation of sensory-motor integration at all postnatal studies. The combination of the 2nd and 3rd issues allows us to merge several techniques like behavioral testing at trail, treadmill, and ladder with invasive procedures just after acomys’s birth. Now we are elaborating some methodological approaches in this sense.

  1. Were the sections heat treated to unblock antigen?

Antigens were unblocked using NaBH4 as in our previous works. This data was added to the manuscript as follows: “Antigens were unmasked in 1% NaBH4 for 15 min (Serva Feinbiochemica, Germany)”.

  1. "For immunohistochemical staining, slices were processed as free floating. Endogenous peroxidase activity was blocked by 0.3% H2O2, unspecific immunoreactivity was lowered by incubation in 3% bovine serum albumin...- how long?

We added this data to the manuscript as follows: “Endogenous peroxidase activity was blocked by 0.3% H2O2 for 30 min, unspecific immunoreactivity was lowered by incubation in 3% bovine serum albumin for 1 h (BSA, Biolot, Russia).

Round 2

Reviewer 2 Report

Comments and Suggestions for Authors

The Authors took my comments into account, and I also thank them for their comprehensive answers.

I believe that the manuscript is suitable for publication in IJMS.